# ChunkAttention: Efficient Attention on KV Cache with Chunking Sharing and Batching

## Abstract

Self-attention is an essential component of GPT-style models and a significant cause of LLM inference latency for long sequences. In multi-tenant LLM inference servers, the compute and memory operation cost of self-attention can be amortized by making use of the probability that sequences from users may share long prompt prefixes. This paper introduces ChunkAttention, a unique self-attention kernel built on chunking, sharing the KV cache, and batching the attention computation. ChunkAttention recognizes matching prompt prefixes across several sequences and shares their KV cache in memory by chunking the KV cache and structuring it into the auxiliary prefix tree. To significantly improve the memory reuse of KV cache and consequently the speed of self-attention for long shared prompts, we design an efficient computation kernel on this new storage structure, where two-phase partitioning is implemented to reduce memory operations on shared KV cache during self-attention. Experiments show that ChunkAttention can speed up self-attention of long shared prompts by 1.6-3x, with lengths ranging from 1024 to 8192.

## 1 Introduction

Over the last few years, Large Language Models (LLMs) have succeeded greatly in many tasks, especially in natural language processing (Chang et al., 2023). By using massive text corpora and strong computing power, researchers have made remarkable progress in training LLMs, and the capabilities of LLMs represented by the GPT, LLaMA, and PaLM series are rapidly improving (OpenAI, 2023; Touvron et al., 2023; Anil et al., 2023).

As LLM applications become widespread, inference cost is a new area of research interest (Kim et al., 2023; Sheng et al., 2023; Aminabadi et al., 2022). Emerging inference frameworks and toolkits include FastTransformer (NVIDIA, 2021), DeepSpeed (Rasley et al., 2020), vLLM (Kwon et al., 2023), and the text-generation-inference server (HuggingFace, 2023). Nevertheless, high inference cost is still the most significant barrier to the commercialization of LLMs. Without aggressive quantization (Frantar et al., 2022; Dettmers et al., 2022), LLMs inference is memory-bound(See Appendix A) for GPUs due to intensive memory operations on KV cache (Williams et al., 2009; Jin et al., 2023). The large KV cache also restricts the batch size. In FP16, KV cache of each token in GPT-3(175B) takes up 4.5M memory, and 2K tokens use up to 9G memory. The memory of an inference server with 8*A100 (80G) can only translate to roughly 70000 tokens or 35 requests. On the other hand, the demand for long sequences is growing fast as LLM applications become complicated. LLMs adapt to this trend by increasing limits on the number of tokens. Up to 32K maximum tokens are supported by GPT-4 (OpenAI, 2023). Ongoing work extends the context window post-training, e.g., position interpolation (Chen et al., 2023). Therefore, reducing the memory and compute cost of self-attention on KV cache, is important for LLM applications.

Due to the high training and deployment cost, LLMs are typically pre-trained and offered to multiple applications in a multi-tenant architecture. For most businesses, it is cost-inefficient for each application to fine-tune models and deploy private instances(single-tenant). To enable the LLMs to gain domain knowledge for specific applications, prompt engineering or in-context learning(ICL) is the key technique for LLM-based applications (Dong et al., 2023; White et al., 2023; Zhou et al., 2023; Brown et al., 2020; Wei et al., 2022). Various LLM applications have revealed that there may be significant overlap in prompts that follow multiple user requests. For example, to build chat services

like ChatGPT and BingChat, we need to add invisible instructions in the prompt to shape the role of the bot and restrict its behavior to meet compliance requirements. These instructions apply to all users equally. User questions are usually much shorter than system prompts. Similarly, LLMs are used by data scientists to annotate data. In this case, the actual corpus to be annotated may only take up a small portion of the prompt because we need to give LLM long and tedious annotation guidelines, which also apply to all requests to LLMs.

Although some work has been done to improve the memory utilization of KV cache (Kwon et al., 2023), there are still unexplored aspects in multi-tenant deployment scenarios where centralized LLM services are provisioned for many application developers. First, no out-of-box solution can automatically discover and remove redundancy in KV cache. Removing redundancy requires service providers and various application developers to establish protocols and manual deployments, which introduces scalability and maintainability issues. Second, more research work needs to be done to fully explore the optimization potential of self-attention algorithms in the case of redundant KV cache. Attention implementations such as xformers (Lefaudeux et al., 2022), FlashAttention (Dao et al., 2022), and PagedAttention (Kwon et al., 2023) do not leverage the shared KV cache to provide extra performance benefits during inference.

To fill the gap, we propose ChunkAttention, a new self-attention kernel built on the KV cache featuring chunking, sharing, and batching. Since sharing happens in prefixes only, after chunking large, continuous key/value tensors into smaller and fragmented ones, we introduce the prefix tree(or forest) as the data structure for chunked KV tensors to discover and remove redundancy at runtime dynamically. Further, ChunkAttention redesigns a high-performance kernel on top of the prefix tree, accelerating self-attention computation and improving the throughput beyond chunking and sharing. This kernel implements two-phase partitioning: chunk-first phase and sequence-first phase. During self-attention computation, query tensors of sequences with matching prompt prefixes are batched together to perform attention with key/value tensors.

The main contributions of this paper are as follows:

- We propose using the prefix tree data structure as the memory management solution for KV cache to remove extra KV cache storage overhead for the scalability and maintainability of multi-tenant LLM services.
- We implement phased partitioning and enable batching to reduce memory operations(MOPs or memory bytes accessed) and further speed up self-attention on KV cache.
- We quantitatively analyze the relationship between self-attention performance under prompt sharing and important hyperparameters, such as shared length, batch size, and hardware specifications.
- Our experiments show that ChunkAttention can achieve comparable performance with the SOTA PagedAttention implementation without prompt sharing and can significantly improve performance with long shared prompts. The specific improvement depends on the length of shared prompts.

## 2 PRELIMINARIES

### 2.1 PROMPT ENGINEERING

Prompt engineering is the practice of developing and optimizing prompts as inputs for LLMs to generate desired outputs. It is widely used in innovative LLM-powered applications, such as question-answering, text summarization, reasoning, and code generation. A typical prompt consists of three parts: instructions, examples, and questions. Instructions serve various purposes, such as shaping roles, providing facts related to the task, and defining policies that LLMs need to follow. Designing good instructions requires domain knowledge and experimentation. Examples are provided as context for LLMs to understand the task, known as in-context learning(ICL) (Dong et al., 2022). Examples are optional for specific tasks since LLMs are trained and tuned on a large amount of data and can perform some tasks in zero-shot settings (Liu et al., 2023). The question is user input data placed at the end of the prompt.

Instructions and examples are usually designed and injected by LLM application developers. They are invisible to users and are identical across multiple user requests. Input data is user-specific and is different.

## 2.2 LLM Inferencing

The typical inference process of LLMs consists of two stages: prefilling and decoding (Sheng et al., 2023). After receiving a sequence $S = [t_1, ..., t_{n_p}]$, the server starts to perform prefilling. During prefilling, it feeds all $n_p$ prompt tokens $t_1, ..., t_{n_p}$ into LLMs simultaneously, computes the attention key/value tensors for all tokens, and caches them to speed up subsequent computations. Then, the server performs decoding. Decoding is auto-regressive, and the input token to LLMs is the completion token(or output token) generated from the previous decoding iteration. The process continues until the end-of-sequence token is generated and $n_c$ completion tokens are generated.

When the server is decoding $b$ (batch size) sequences $S_1, ..., S_b$ simultaneously, although they are in different iterations, the server can still perform batching at the granularity of iteration and predict the next tokens for all sequences together, rather than separately, which is known as iteration-based batching (Gao et al., 2018; Yu et al., 2022; Silfa et al., 2022). Specifically, iteration-based batching concatenates last input tokens of multiple sequences(one token per sequence) $t^{(1)}, ..., t^{(b)}(t^{(i)} \in S_i)$ into a single input $T$, and computes the QKV linear layer before self-attention, the fully connected layer and the projection layer after self-attention. The self-attention in the middle has no shared weights and needs to be computed independently for each sequence. During decoding, new sequences can join, and completed sequences can leave, significantly increasing the possibility of forming big batches. The ChunkAttention in this paper assumes that iteration-based batching is enabled to form batches for its kernel to run efficiently.

## 2.3 Chunking of KV Cache

In various self-attention implementations, KV cache is stored in dense tensors of size $b \times h \times n \times d$ where $b$ is the batch size, $h$ is the number of heads, $n$ is the sequence length and $d$ is the head dimension size. During decoding, only one token is generated per iteration, and its key/value tensors need to be concatenated into KV cache of previous tokens to continue decoding. To avoid resizing, the common practice is pre-allocating a contiguous memory buffer of maximum possible sequence length $n_{max}$. It leads to significant memory waste when the actual token count is much less than $n_{max}$.

The chunking(or paging) technique breaks big KV cache contiguous in memory into a list of smaller chunks along the sequence length dimension (Kwon et al., 2023). Each chunk stores part of the key/value tensors of size $h \times c \times d$, where the chunk size $c$ is defined by the number of tokens covered by each chunk. Chunking changes the memory allocation of KV cache from pre-allocation to on-demand allocation to reduce memory waste. To guarantee equal chunk size, some memory space for alignment is unused. Given that the sequence length is $n$, the memory loss is bounded by $(c - 1)/n$.

## 3 Implementation of ChunkAttention

### 3.1 Chunking and Sharing of KV Cache in Prefix Tree

When two or more sequences share a series of common prefix tokens in prompts (usually instructions and examples for prompt engineering), KV cache of prefix tokens is the same and thus can be shared in memory. For example, a particular LLM inference server receives sequence $S_i = [t_1, ..., t_{n_s}, t_{n_s+1}, ..., t_{n_p}]$ first, and then receives sequence $S_j = [t_1, ..., t_{n_s}, t'_{n_s+1}, ..., t'_{n_p}]$. KV cache for $t_1, ..., t_{n_s}$ can only have one physical copy in memory.

We suggest using this property and organizing chunks of various sequences in a prefix tree(or a trie) to detect and eliminate the extra memory footprint of KV cache at runtime. Figure 1 shows an overall structure of the KV cache stored in a prefix tree. Each node of the tree is defined by a chunk $C$ storing three essential elements: i) context tokens $t_{m+1}, ..., t_{m+c}$ shared by sequences $S_i, ..., S_j$, or a hash for big chunk size, to enable prefix tree operations; ii) the key tensor; ii) the value tensor. Each path of the prefix tree corresponds to a sequence.

During inference, there are three possible scenarios: new sequence joins, completed sequence leaves, and all sequences decode once at one iteration. Each scenario can be translated into operations on the prefix tree. When a new sequence joins, the prefix tree is searched and updated to include a new

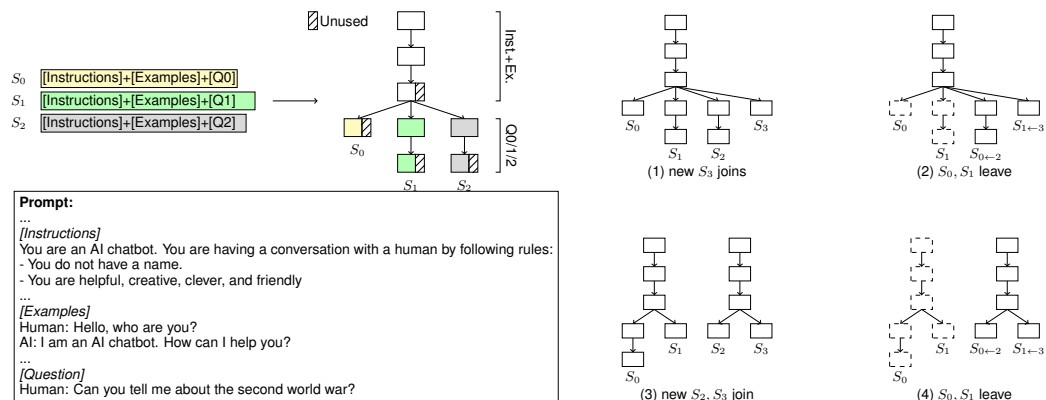

Figure 1: KV cache in prefix tree. The instructions and examples in prompts of $S_0, S_1, S_2$ are common and sharable. Questions are different and not sharable. Some memory is unused due to alignment. As new sequences join and old sequences leave, sequence indexes are adjusted to maintain continuity.

path and leaf node to store its unique KV cache. When a completed sequence leaves, the prefix tree is updated to delete its corresponding KV cache. At each decoding iteration, we check whether the chunk at the leaf node is full and decide whether to grow new leaf nodes.

Depending on specific sharing scenarios, multiple trees (a forest) may exist in the server simultaneously. For instance, different prompts are created by various LLM application developers.

The parent-child relationship defines the subset of sequences that each chunk covers. The root node covers all sequences, and the leaf nodes cover only one sequence. A key property of the prefix tree is that sequences covered by each chunk in a prefix tree or forest are contiguous in the sequence index dimension. Therefore, slicing the query tensor in self-attention is particularly efficient during kernel computation, which will be discussed in more detail in the next section.

The prefix tree structure is a natural fit for the CPU kernel, but for the GPU kernel, we must synchronize metadata, such as the chunk list and its covered sequence start and end indexes, between CPU and GPU memory. For example, in Figure 2, we need to pass $(C_0/C_1/C_2, 0, 2)$, $(C_3, 0, 0)$, $(C_4/C_6, 1, 1)$, and $(C_5/C_7, 2, 2)$. There is some overhead as a result. However, the prefix tree does not change at every decoding iteration, and we can cache the metadata in GPU memory and only synchronize incremental changes when the tree structure changes. The events that trigger synchronization are chunk full, new sequence join, and completed sequence leave. These overheads can be successfully amortized and masked by multiple rounds of iterations.

Given a fixed chunk size, memory management via a prefix tree is efficient. In ChunkAttention, the pool-based memory allocator is adopted by default (Trebino, 2016). It keeps track of both a used and a free chunk list. When a new chunk is requested, the allocator returns a chunk from the free list or allocates fresh memory from the operating system (OS). Unused chunks are returned to the allocator once a sequence is completed, but the allocator does not release memory to the OS, preventing additional memory requests.

The chunk allocator interface can be implemented using a different memory management solution since it is not coupled to the upper-level self-attention kernel algorithm discussed in the next section. In contrast to the above on-demand technique, we can, for instance, pre-allocate all chunks during service start-up (ahead of time) to speed up early chunk allocations or even use a variable chunk size.

By sharing common prefixes, the number of sequences that can be processed simultaneously is increased by approximately $1/(1 - r)$. The sharing ratio $r$ is defined by the percentage of shared prefix tokens $n_s/(n_p + n_c)$. In memory-limited inference scenarios, this helps increase the batch size and thus improve throughput.

Although we focus on sharing prompt prefixes across user requests in this paper, the proposed solution is compatible with various scenarios where KV cache can be compressed. For example, $S_i$ and

$S_j$ are multiple results of a single user request, and they share all prompt tokens. Or $S_i$ beats other candidates and turns into multiple sequences $S_i$ and $S_{i+1}$ during sampling in a particular decoding iteration. However, sharing prompt prefixes has the most memory-saving potential in practice and is our focus in this paper.

## 3.2 EFFICIENT KERNEL WITH PHASED PARTITIONING AND BATCHING

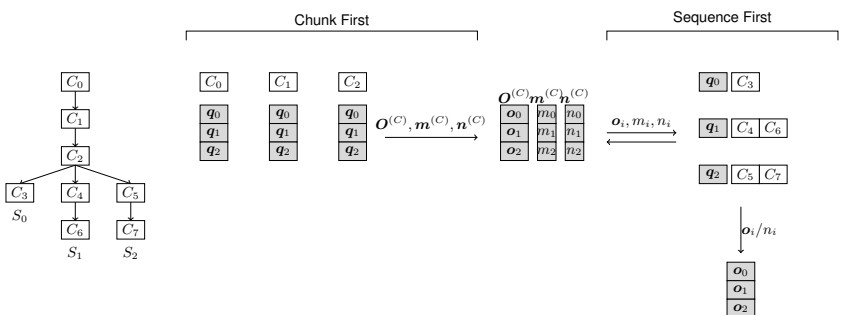

Figure 2: Self-attention kernel of ChunkAttention. The server is decoding sequences $S_0$, $S_1$, and $S_2$. They share chunks $C_0$, $C_1$ and $C_2$. In the chunk-first phase, queries $q_0$, $q_1$ and $q_2$ are batched for self-attention with $C_0$, $C_1$ and $C_2$. Partial attention result $O^{(C)}$, $m^{(C)}$ and $n^{(C)}$ are saved into memory. In the sequence-first phase, $o_i$, $m_i$, and $n_i$ for each sequence are restored, and we continue processing the remaining chunks with respect to $q_i$ only.

---

**Algorithm 1** Self Attention: Chunk First (partition chunks)

---

**Require:** $Q \in \mathbb{R}^{b \times d}$ (query), $T$(prefix tree)
**Ensure:** $O \in \mathbb{R}^{b \times d}$ (attention output)
1: **function** ATTNCHUNKFIRST($Q$, $T$)
2:     Get chunks $C_1, ..., C_k$ in $T$ that are shared by multiple sequences
3:     $O, m, n \leftarrow 0, 0, 0$
4:     **for** $C \leftarrow C_1$ to $C_k$ **do**
5:         $K^{(C)}, V^{(C)} \leftarrow$ key, value cache stored in $C$
6:         $i, j \leftarrow$ start index, end index of sequences covered by $C$
7:         $O^{(C)}, m^{(C)}, n^{(C)} \leftarrow \textbf{\textit{partial\_attn}}(Q, K^{(C)}, V^{(C)}, i, j)$
8:         Save partial attention result $O^{(C)}, m^{(C)}, n^{(C)}$ to memory
9:     **end for**
10: **end function**

---

In this section, we dive into the self-attention kernel implementation in ChunkAttention. It is built on top of the unique data storage that KV cache of each sequence spans multiple non-contiguous memory chunks in a prefix tree, and some chunks are shared. During prefilling, we can perform a lookup to avoid re-computation of QKV projections for shared chunks before applying any existing highly optimized self-attention kernels on prompts, e.g., FlashAttention (Dao, 2023). Then key/value tensors are chunked and saved into the prefix tree.

During decoding iterations of ChunkAttention, the self-attention computation is divided into chunk-first and sequence-first phases. The two phases focus on different queries, KV cache chunks, and parallelization strategies. We adopt the online softmax algorithm to reduce the synchronization requirement between partitions (Milakov & Gimelshein, 2018; Dao, 2023). The overall process is shown in Figure 2. Since heads are always partitioned, they are omitted in the following discussion.

**Chunk-first Phase**. In the chunk-first phase, we only process those chunks shared by multiple sequences. Since the number of GPU streaming multiprocessors (108 SMs for A100) is usually much bigger than the number of heads (32 for Llama-7B), partitioning only by heads can only partially utilize the hardware resources. The number of query tokens in decoding is always 1, which limits the possibility of partitioning on queries. We need to perform additional partitioning on keys/values. Chunking itself provides a convenience for partitioning keys/values.

The computation is performed by traversing chunks in the prefix tree and executing the partial attention kernel *partial_attn* on them and saving the partial attention results into memory, as shown in Algorithm 1. $b$ and $h$ are the number of sequences (batch size) in the server and the number of heads. $Q \in \mathbb{R}^{b \times d}$ is the queries formed by concatenating the last tokens of all $b$ sequences in the latest decoding iteration.

The operations in *partial_attn* are given by Equation 1. It computes the partial attention result $O^{(C)}$, $m^{(C)}$ and $n^{(C)}$ with respect to each chunk $C$ independently, thus it can be parallelized. $Q_{i:j,:}$ is a slice of $Q$ for sequences ranging from $i$ to $j$ which share the KV cache stored in Chunk $C$. The maximum attention weights vector $M^{(C)}$ is the row-wise max over the last dimension of attention weights $W^{(C)}$. The softmax normalization term $n^{(C)}$ is the row-wise sum over the last dimension of $E^{(C)}$. $M^{(C)}$ and $n^{(C)}$ are auxiliary variables introduced to further cumulate partial attention results of multiple chunks.

The *partial_attn* efficiently accesses shared KV cache memory as self-attentions for multiple sequences are batched. It happens at a granularity of dot-product between queries $Q_{i,:}, ..., Q_{j,:}$ of sequences $S_i, ..., S_j$ and shared $K^{(C)}/V^{(C)}$. In addition to reduced memory access, another benefit of batching is to turn the query from a vector into a matrix, allowing efficient matrix multiplications with tensor cores.

$$
\begin{aligned}
W^{(C)} &= Q_{i:j,:}K^{(C)} \in \mathbb{R}^{(j-i) \times c} \\
m^{(C)} &= \max\left(W^{(C)}\right) \in \mathbb{R}^{(j-i)} \\
E^{(C)} &= \exp\left(W^{(C)} - m^{(C)} \cdot \mathbf{1}^T\right) \in \mathbb{R}^{(j-i) \times c} \\
n^{(C)} &= \operatorname{sum}\left(E^{(C)}\right) \in \mathbb{R}^{(j-i)} \\
O^{(C)} &= E^{(C)}V^{(C)} \in \mathbb{R}^{(j-i) \times d}
\end{aligned}
\tag{1}
$$

**Sequence-first Phase**. In the sequence-first phase, we load partial attention results of shared chunks from the chunk-first phase and continue processing those chunks only related to the current sequence. We partition sequences. Each $q$ handled by the sequence-first kernel is a vector by slicing the $i$-th row of $Q$, as shown in Algorithm 2.

---

**Algorithm 2** Self Attention: Sequence First (partition sequences)

---

**Require:** $Q \in \mathbb{R}^{b \times d}$ (query), $T$(prefix tree)
**Ensure:** $O \in \mathbb{R}^{b \times d}$ (attention output)
1: **function** ATTNSEQFIRST($Q, T$)
2:     **for** $q \leftarrow q_1$ to $q_b$ **do**
3:         $o, m, n \leftarrow \mathbf{0}, 0, 0$
4:         Get partial attention results $\left(O^{(C_1)}, m^{(C_1)}, n^{(C_1)}\right), ..., \left(O^{(C_k)}, m^{(C_k)}, n^{(C_k)}\right)$
5:         **for** $O^{(C)}, m^{(C)}, n^{(C)} \leftarrow \left(O^{(C_1)}, m^{(C_1)}, n^{(C_1)}\right)$ to $\left(O^{(C_k)}, m^{(C_k)}, n^{(C_k)}\right)$ **do**
6:             Partial attention result of $q$: $o^{(C)}, m^{(C)}, n^{(C)} \leftarrow$ slicing $O^{(C)}, m^{(C)}, n^{(C)}$
7:             ***attn_reduce***$(o^{(C)}, m^{(C)}, n^{(C)}, o, m, n)$
8:         **end for**
9:         Get chunks $C_{k+1}, C_{k+2}..., C_l$ in $T$ with respect to $q$ only
10:         **for** $C \leftarrow C_{k+1}$ to $C_l$ **do**
11:             $K^{(C)}, V^{(C)} \leftarrow$ key, value cache stored in $C$
12:             $i \leftarrow$ sequence index of $q$
13:             $o^{(C)}, m^{(C)}, n^{(C)} \leftarrow$ ***partial_attn***$(q, K^{(C)}, V^{(C)}, i, i+1)$
14:             ***attn_reduce***$(o^{(C)}, m^{(C)}, n^{(C)}, o, m, n)$
15:         **end for**
16:     **end for**
17: **end function**

---

The *attn_reduce* repeatedly merges partial attention result of one chunk, $o^{(C)}, m^{(C)}, n^{(C)}$, produced by the *partial_attn* into the cumulative attention result $o$, $m$, and $n$ by scaling them with $x^{(C)}$ and

$y^{(C)}$ respectively. Equation 2 shows the process. $\boldsymbol{O}_{i,:}$, $\boldsymbol{m}_i$ and $\boldsymbol{n}_i$ are slices for sequence of index $i$. The final attention output is given by $\boldsymbol{O}/\boldsymbol{n}$ element-wise.

$$
\begin{aligned}
x^{(C)} &= \exp\left(m^{(C)} - \max\left(m^{(C)}, \boldsymbol{m}_i\right)\right) \in \mathbb{R} \\
y^{(C)} &= \exp\left(\boldsymbol{m}_i - \max\left(m^{(C)}, \boldsymbol{m}_i\right)\right) \in \mathbb{R} \\
\boldsymbol{O}_{i,:} &= x^{(C)}\boldsymbol{o}^{(C)} + y^{(C)}\boldsymbol{O}_{i,:} \in \mathbb{R}^d \\
\boldsymbol{n}_i &= x^{(C)}n^{(C)} + y^{(C)}\boldsymbol{n}_i \in \mathbb{R} \\
\boldsymbol{m}_i &= \max\left(m^{(C)}, \boldsymbol{m}_i\right) \in \mathbb{R}
\end{aligned}
\tag{2}
$$

The sequence-first phase is efficient in terms of concurrency, as *partial_attn* and *attn_reduce* are performed locally, without communication between thread blocks. However, without the partial attention results from the chunk-first phase, it needs to read shared chunks $b$ times, which adds significant memory IO overhead. With a large hardware cache(L2 cache of A100 is 80 MB), the problem can be alleviated, though not eliminated. We achieve the trade-off between them with two-phase partitioning, and it applies to both the CPU and CPU devices. Appendix B gives the memory operations analysis of different partition strategies.

The temporary memory allocated to store partial attention results in chunk-first phase can be eliminated when atomic operations are not a bottleneck, e.g., on CPU devices. The *attn_reduce* can be executed right after *partial_attn* in chunk-first phase to directly merge partial attention results into the final result. Since multiple shared chunks with a parent-child relationship in the prefix tree write into the same slice of $\boldsymbol{O}, \boldsymbol{m}, \boldsymbol{n}$, reduce needs to be serialized. By default, we avoid the overhead by writing partial attention results of different shared chunks into their independent temporary memory (size $b \times d$ per chunk), and the reduce function is deferred to the sequence-first phase. In the CPU kernel, the overhead of serializing reduce is insignificant compared to computation and reduce can be implemented using spin locks to avoid temporary memory management.

## 4  RELATED WORK

The most relevant work on reducing the memory footprint of KV cache is PagedAttention in vLLM (Kwon et al., 2023). It introduced the paging technique in OS to solve the problem of memory waste caused by dynamic and unknown sequence lengths during LLM decoding. However, sharing between user requests is not implemented by vLLM releases and must instead be enabled manually after considerable effort. Our solution, which differs from the paging one, uses the prefix tree to manage memory and aims to discover redundant KV cache across user requests at runtime automatically. The solution is more practical for multi-tenant deployment scenarios where LLM service providers centrally host models and have high scalability and maintainability requirements. According to vLLM, the shared KV cache is similar to the dynamic link library shared by multiple processes in the operating system. As opposed to PagedAttention's strategy of compiling and publishing it in advance (AoT), we expect compiling and loading it in real-time (JIT). Additionally, to the best of our knowledge, no method has been proposed to explore the optimization opportunities brought by shared KV cache. Based on the metadata captured by the prefix tree, our work further explores this problem. It proposes two-phase partitioning to improve performance, which is another difference between our work and the existing work.

Partition strategies in ChunkAttention are built on online softmax (Milakov & Gimelshein, 2018) and inspired by FlashAttention (Dao et al., 2022; Dao, 2023), which adopted the same algorithm. FlashAttention thoroughly researched and implemented various tiling techniques, accelerating attention by 2-4x while cutting memory operations by 10-20x. FlashAttention-2 altered tiling strategies and additionally doubled the speed. However, FlashAttention is inflexible regarding non-contiguous memory or variable sequence lengths, making it more suitable for model training or prefilling of model inference than iterative decoding in multi-tenant servers. Moreover, there is little gain when the query token count is always one during decoding. ChunkAttention handles variable sequence lengths during decoding and batches attention operations of several sequences to reduce memory operations. As a result, our work and FlashAttention are complementary.

## 5    EXPERIMENTS

**Hardware**. We primarily use the NVIDIA A100(80G) for evaluation. Since inference scenarios are more diverse than training and require less bandwidth or compute resources, we also run benchmarks on A100(40G) and RTX 4090, capable of running models of certain sizes.

**Baselines**. We select four high-performance self-attention implementations as baselines: Naive PyTorch implementation by the formula $\mathrm{softmax}(\boldsymbol{Q}\boldsymbol{K}^T/\sqrt{d})\boldsymbol{V}$, the memory-efficient self-attention implemented in xformers (Lefaudeux et al., 2022), FlashAttention integrated in PyTorch (Dao et al., 2022), and PagedAttention in vLLM (Kwon et al., 2023). Naive, xformers, and FlashAttention do not implement sharing. For PagedAttention, sharing cannot be achieved automatically at runtime, and we manually construct a paging memory area with shared KV cache to simulate sharing enabled. None of them support batching.

**Implementation**. We implement and compile the ChunkAttention GPU kernel with CUDA 11.8 on A100 and CUDA 12.0 on RTX 4090. Iteration-based batching is implemented, and the sequence lengths can be different.

**Workload**. Sequences are processed in batch mode. All sequences within the same batch start and finish simultaneously. Each sequence is prefilled with $n_s$ shared prompt tokens, followed by 64 random tokens to make them start to diverge. Thus, the total prompt token count is $n_p = n_s + 64$. The task is to decode the next $n_c$ completion tokens iteratively. We measure the elapsed decoding time $t$ and the throughput defined by token rate(tokens per second or tps, $n_c * b/t$). For all experiments, the head dimension $d$ is 128, the number of heads $h$ is 32, and the chunk size is fixed to be 64. All tensors are in FP16.

**Results**. We run experiments to observe the performance improvement by KV cache sharing and batching from three aspects: number of shared tokens, batch size, and different hardware, each with a group of experiments. The first two groups of experiments run on A100(80G) only.

Figure 3 focuses on shared token count. PagedAttention with sharing is about 2x faster than Naive PyTorch for a shared token count from 256 to 8192. ChunkAttention performs as well as the SOTA PagedAttention when the shared token count is lower than 512. When the shared token count is higher than 1024, ChunkAttention significantly outperforms PagedAttention. When $n_s$ reaches 8192, ChunkAttention yields 3x throughput improvement, resulting from the two-phase partitioning and fine-grained batching. For all configurations, the throughput decreases as the decoding proceeds. However, when $n_s$ reaches 8192, the performance advantage of ChunkAttention is still obvious after generating 1024 tokens.

Figure 4 focuses on the batch size. For those implementations without sharing implemented, the token rate is almost agnostic to batch size due to memory-bound(See Appendix A). For ChunkAttention and PagedAttention, the token rate peaks when the batch size is 48 or 64. Batching can only be effective when sharing is enabled in self-attention kernels, as we do in ChunkAttention.

Figure 5 shows results on different hardware. We remove attention implementations without sharing implemented and focus on ChunkAttention and PagedAttention. ChunkAttention has more obvious acceleration on all hardware, especially for long shared prompts. We can see that on both A100(40G) and A100(80G), ChunkAttention's performance advantage appears when the shared token count reaches 512. RTX 4090 performs better than A100(80G), and the obvious batching gain only appears when the shared token count reaches 2048.

## 6    CONCLUSION

In this paper, we propose ChunkAttention, a novel self-attention kernel, to accelerate self-attention for LLM inference. On top of the chunking technique, we introduce the prefix tree to manage the chunked KV cache. It addresses the challenge of detecting and removing redundant KV cache at runtime in a multi-tenant environment. We evaluated ChunkAttention in various configurations and hardware. It showed that ChunkAttention can achieve comparable throughput with SOTA PagedAttention with short shared prompt prefixes and can outperform it by 3x with a long shared prompt prefix of 8192 tokens on A100(80G) by applying two-phase partitioning and enabling batching. As hardware evolves, the effect of batching will only appear with longer shared prompt prefixes.

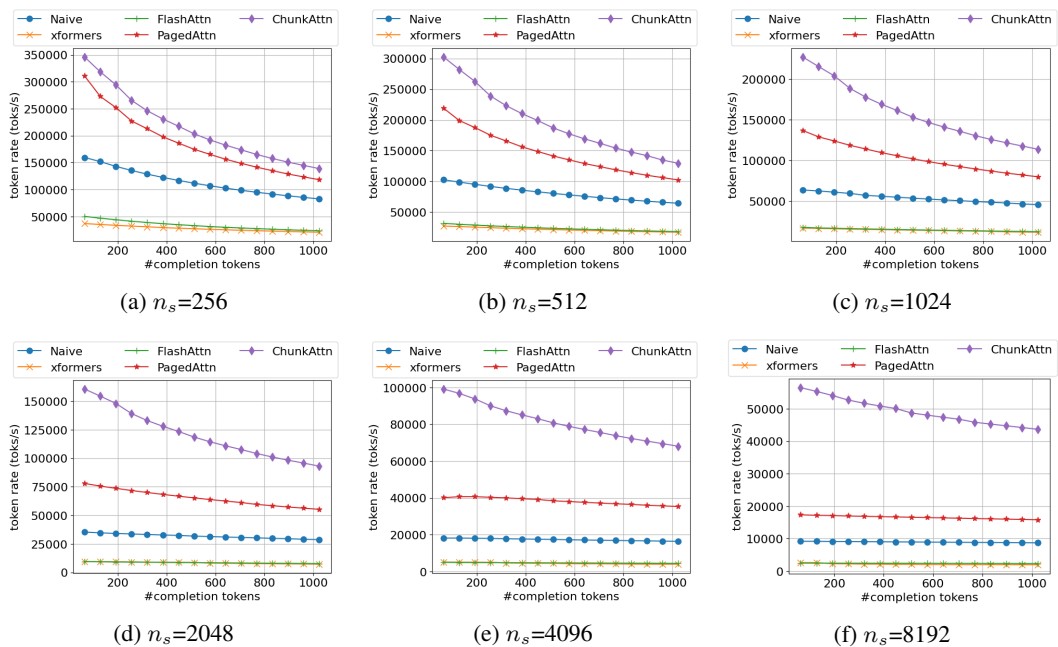

Figure 3: Token rate of decoding $n_c$ completion tokens given various shared token counts(A100 80G, FP16). Chunk size $c$=64, batch size $b$=32.

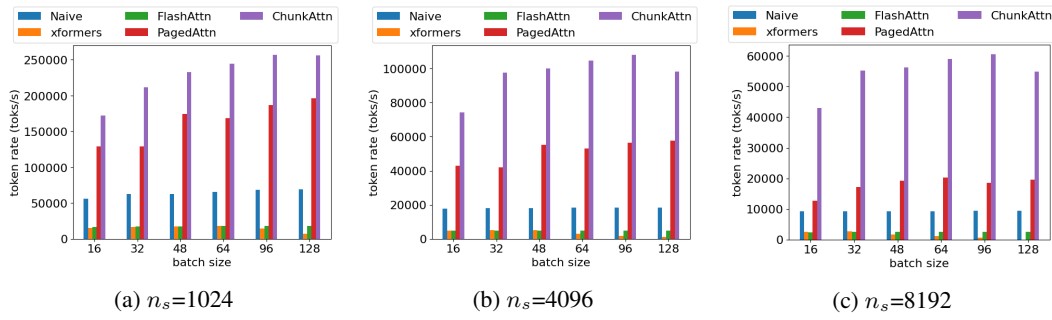

Figure 4: Token rate of decoding $n_c$=64 completion tokens at various batch sizes(A100 80G, FP16). Chunk size $c$=64.

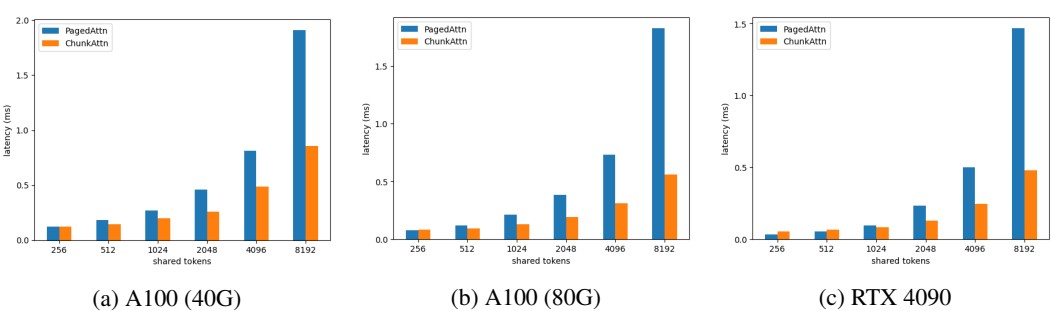

Figure 5: Latency of decoding the first completion token on different hardware(FP16). Chunk size $c$=64, batch size $b$=32.

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

# A  ARITHMETIC INTENSITY ANALYSIS

Table 1: Arithmetic Intensity at the Prefill Stage for llama2 7B with Batch Size = 1

| Prompt Length | Roofline | QKV Projection | softmax $\left(\frac{QK^T}{\sqrt{d_k}}\right)V$ | MLP | Output Projection |
|---|---|---|---|---|---|
| 1024 | FLOPs($\times 10^9$)
IO($\times 10^9$)
Intensity | 103.08
0.13
768.00 | 17.18
0.17
102.40 | 277.03
0.36
762.46 | 268.44
0.34
798.75 |
| 2048 | FLOPs($\times 10^9$)
IO($\times 10^9$)
Intensity | 206.16
0.17
1228.80 | 68.72
0.60
113.78 | 554.05
0.46
1214.68 | 536.87
0.41
1309.46 |
| 3072 | FLOPs($\times 10^9$)
IO($\times 10^9$)
Intensity | 309.24
0.20
1536.00 | 154.62
1.31
118.15 | 831.08
0.55
1513.99 | 805.31
0.48
1664.14 |
| 4096 | FLOPs($\times 10^9$)
IO($\times 10^9$)
Intensity | 412.32
0.23
1755.43 | 274.88
2.28
120.47 | 1108.10
0.64
1726.75 | 1073.74
0.56
1924.81 |
| 5120 | FLOPs($\times 10^9$)
IO($\times 10^9$)
Intensity | 515.40
0.27
1920.00 | 429.50
3.52
121.90 | 1385.13
0.73
1885.74 | 1342.18
0.63
2124.48 |
| 6144 | FLOPs($\times 10^9$)
IO($\times 10^9$)
Intensity | 618.48
0.30
2048.00 | 618.48
5.03
122.88 | 1662.15
0.83
2009.06 | 1610.61
0.71
2282.32 |
| 7168 | FLOPs($\times 10^9$)
IO($\times 10^9$)
Intensity | 721.55
0.34
2150.40 | 841.81
6.81
123.59 | 1939.18
0.92
2107.51 | 1879.05
0.78
2410.22 |
| 8192 | FLOPs($\times 10^9$)
IO($\times 10^9$)
Intensity | 824.63
0.37
2234.18 | 1099.51
8.86
124.12 | 2216.20
1.01
2187.93 | 2147.48
0.85
2515.97 |

Table 2: Arithmetic Intensity at the Prefill Stage for llama2 7B with Prompt Length = 1024

| Batch Size | Roofline | QKV Projection | softmax $\left(\frac{QK^T}{\sqrt{d_k}}\right) V$ | MLP | Output Projection |
|---|---|---|---|---|---|
| 1 | FLOPs($\times 10^9$) | 103.08 | 17.18 | 277.03 | 268.44 |
| | IO($\times 10^9$) | 0.13 | 0.17 | 0.36 | 0.34 |
| | Intensity | 768.00 | 102.40 | 762.46 | 798.75 |
| 4 | FLOPs($\times 10^9$) | 412.32 | 68.72 | 1108.10 | 1073.74 |
| | IO($\times 10^9$) | 0.23 | 0.67 | 0.64 | 0.56 |
| | Intensity | 1755.43 | 102.40 | 1726.75 | 1924.81 |
| 8 | FLOPs($\times 10^9$) | 824.63 | 137.44 | 2216.20 | 2147.48 |
| | IO($\times 10^9$) | 0.37 | 1.34 | 1.01 | 0.85 |
| | Intensity | 2234.18 | 102.40 | 2187.93 | 2515.97 |
| 16 | FLOPs($\times 10^9$) | 1649.27 | 274.88 | 4432.41 | 4294.97 |
| | IO($\times 10^9$) | 0.64 | 2.68 | 1.76 | 1.44 |
| | Intensity | 2586.95 | 102.40 | 2525.13 | 2972.42 |
| 32 | FLOPs($\times 10^9$) | 3298.53 | 549.76 | 8864.81 | 8589.93 |
| | IO($\times 10^9$) | 1.17 | 5.37 | 3.24 | 2.63 |
| | Intensity | 2808.69 | 102.40 | 2735.97 | 3268.95 |
| 64 | FLOPs($\times 10^9$) | 6597.07 | 1099.51 | 17729.62 | 17179.87 |
| | IO($\times 10^9$) | 2.25 | 10.74 | 6.21 | 4.99 |
| | Intensity | 2934.45 | 102.40 | 2855.17 | 3440.57 |
| 96 | FLOPs($\times 10^9$) | 9895.60 | 1649.27 | 26594.44 | 25769.80 |
| | IO($\times 10^9$) | 3.32 | 16.11 | 9.18 | 7.36 |
| | Intensity | 2978.91 | 102.40 | 2897.24 | 3501.85 |
| 128 | FLOPs($\times 10^9$) | 13194.14 | 2199.02 | 35459.25 | 34359.74 |
| | IO($\times 10^9$) | 4.40 | 21.47 | 12.15 | 9.72 |
| | Intensity | 3001.65 | 102.40 | 2918.74 | 3533.32 |

Table 3: Arithmetic Intensity at the Decoding Stage for llama2 7B

| Batch Size | Context Length | QKV Projection | softmax $\left(\frac{QK^T}{\sqrt{d_k}}\right) V$ | MLP | Output Projection |
|---|---|---|---|---|---|
| 1 | 2048 | 0.99967 | 0.99177 | 0.99967 | 0.99972 |
| | 4096 | 0.99967 | 0.99201 | 0.99967 | 0.99972 |
| | 6144 | 0.99967 | 0.99209 | 0.99967 | 0.99972 |
| | 8192 | 0.99967 | 0.99213 | 0.99967 | 0.99972 |
| 4 | 2048 | 3.99480 | 0.99177 | 3.99465 | 3.99560 |
| | 4096 | 3.99480 | 0.99201 | 3.99465 | 3.99560 |
| | 6144 | 3.99480 | 0.99209 | 3.99465 | 3.99560 |
| | 8192 | 3.99480 | 0.99213 | 3.99465 | 3.99560 |
| 8 | 2048 | 7.97922 | 0.99177 | 7.97862 | 7.98241 |
| | 4096 | 7.97922 | 0.99201 | 7.97862 | 7.98241 |
| | 6144 | 7.97922 | 0.99209 | 7.97862 | 7.98241 |
| | 8192 | 7.97922 | 0.99213 | 7.97862 | 7.98241 |
| 16 | 2048 | 15.91710 | 0.99177 | 15.91470 | 15.92981 |
| | 4096 | 15.91710 | 0.99201 | 15.91470 | 15.92981 |
| | 6144 | 15.91710 | 0.99209 | 15.91470 | 15.92981 |
| | 8192 | 15.91710 | 0.99213 | 15.91470 | 15.92981 |
| 32 | 2048 | 31.67010 | 0.99177 | 31.66061 | 31.72046 |
| | 4096 | 31.67010 | 0.99201 | 31.66061 | 31.72046 |
| | 6144 | 31.67010 | 0.99209 | 31.66061 | 31.72046 |
| | 8192 | 31.67010 | 0.99213 | 31.66061 | 31.72046 |
| 64 | 2048 | 62.69388 | 0.99177 | 62.65671 | 62.89154 |
| | 4096 | 62.69388 | 0.99201 | 62.65671 | 62.89154 |
| | 6144 | 62.69388 | 0.99209 | 62.65671 | 62.89154 |
| | 8192 | 62.69388 | 0.99213 | 62.65671 | 62.89154 |
| 96 | 2048 | 93.09091 | 0.99177 | 93.00898 | 93.52737 |
| | 4096 | 93.09091 | 0.99201 | 93.00898 | 93.52737 |
| | 6144 | 93.09091 | 0.99209 | 93.00898 | 93.52737 |
| | 8192 | 93.09091 | 0.99213 | 93.00898 | 93.52737 |
| 128 | 2048 | 122.88000 | 0.99177 | 122.73728 | 123.64163 |
| | 4096 | 122.88000 | 0.99201 | 122.73728 | 123.64163 |
| | 6144 | 122.88000 | 0.99209 | 122.73728 | 123.64163 |
| | 8192 | 122.88000 | 0.99213 | 122.73728 | 123.64163 |

# B IO ANALYSIS OF CHUNKATTENTION KERNEL

$b$ is the batch size. $c$ is the chunk size. $d$ is the head dimension size. $h$ is the number of heads. $n$ is the sequence length. $r$ is the ratio of the shared prompt length to the total sequence length. Thus, the number of shared chunks is given by $nr/c$.

Memory operations of ChunkAttention kernel:

$$\begin{aligned}
\text{MOPs} &= h(2cd + 2bd)\frac{nr}{c} + 2hb(d + n(1-r)d + d) \\
&= bhd\left(2nr(\frac{1}{b} + \frac{1}{c}) + 2 + 2n(1-r)\right)
\end{aligned} \tag{3}$$

Memory operations of naive kernel:

$$\begin{aligned}
\text{MOPs} &= bh(2d + 2nd) \\
&= bhd(2 + 2n)
\end{aligned} \tag{4}$$

Typically, the chunk size $c$ is greater than 32. If we have a batch size of $n \geq 2$, the following condition is easy to meet. Then, the ChunkAttention kernel has fewer memory operations than the naive kernel.

$$\frac{1}{b} + \frac{1}{c} < 1 \tag{5}$$

