# OpenReview forum: "ChunkAttention: Efficient Attention on KV Cache with Chunking Sharing and Batching"
_ICLR.cc/2024/Conference — Submitted to ICLR 2024_

### Official Review · Reviewer_oC3j · 2023-10-22

**Soundness:** 2 fair
**Presentation:** 2 fair
**Contribution:** 3 good
**Rating:** 3
**Confidence:** 5

**Summary:**

For auto-regressive large language models (LLMs), efficient attention contributes to cost-efficient serving. Inspired by the "prompt engineering" of LLM inference, this paper proposes a novel attention mechanism, named ChunkAttention. Since all requests to an LLM model may share the predefined prompt prefix, the authors build a prefix tree and match the common prefix. The common prefix can be computed only once, and all requests can share the computed key-value tensors. This results in saving both computation (no duplicate computation) and memory (no duplicate saving) thus enhancing the efficiency of online serving.

**Strengths:**

- The authors found a good optimization point from practical LLM serving scenarios.
- The authors adopted a plausible abstraction--prefix tree--to realize their idea.
- Figure 5 shows their contribution well, showing we can dramatically decrease the prompt encoding time.

**Weaknesses:**

- The authors said that in their experiment workloads, "all sequences within the same batch start and finish simultaneously". However, I think the workload is overfitted to ChunkAttention. It seems more natural to send a trace of requests through a fixed time duration.
    - For example, it would be helpful to understand ChunkAttention's contribution if I could compare under a similar workload to Figure 16 of the vLLM paper
- Gathering that many batches (>= 32) is unlikely in practice. Even for a super high load scenario, requests would suffer from queueing delay, failing to meet SLA.
- It would be more informative if a study about the chunk size existed.

**Questions:**

- In Figure 4, why does the throughput decrease when the batch size becomes 128?
- Although 1) naive, 2) xformers, and 3) FlashAttention do not support sharing, why does the naive version outperform the others?
- When the model size grows on the same hardware (e.g., Lllama 13B in A100 80G), how does the result change? I think ChunkAttention can achieve more performance gain because it efficiently utilizes key-value cache memory compared to the baselines.
- How long does it take to build the first prefix tree? For example, when $n_{s} = 8192$, I think it might take a non-negligible time.

---

> ### Author Response · Authors · 2023-11-18
>
> > Question 1: In Figure 4, why does the throughput decrease when the batch size becomes 128?
>
> - Regarding the ChunkAttn implementation, it encompasses two CUDA kernel functions: chunk_first and seq_first.
> - In the case of seq_first, optimal GPU utilization occurs when attention_head_size multiplied by batch_size exceeds the GPU's streaming multiprocessor count. For head 32, when the batch size is greater than 48, the throughput stabilizes.
> - As for chunk_first, the kernel's performance primarily hinges on matrix multiplication. In theory, larger batches should yield improved throughput. However, our matrix multiplication implementation lacks fine-tuning for batch size 128, resulting in this regression.
>
> > Question 2: Although 1) naive, 2) xformers, and 3) FlashAttention do not support sharing, why does the naive version outperform the others?
>
> - xformers and FlashAttention focus on training instead of inference, which means they lack specifical design for generation: the Q in QKV is a vector instead matrix.
> - Flash attention has been aware of this issue and published Flash-decoding. From its [blog](https://pytorch.org/blog/flash-decoding/), we also could find the flash attention outperforms naive pytorch(figure3).
>
> > Question 3: When the model size grows on the same hardware (e.g., Lllama 13B in A100 80G), how does the result change? I think ChunkAttention can achieve more performance gain because it efficiently utilizes key-value cache memory compared to the baselines.
>
> - I believe ChunkAttention can yield greater performance gains due to its efficient utilization of key-value cache memory compared to the baseline methods. When the model size increases, ChunkAttention demonstrates improved performance for two reasons: 1. Enhanced utilization of L1/L2 cache, and 2. Memory savings from sharing that allow for larger acceptable batch sizes.
> - For comprehensive end-to-end testing of the entire model, more time might be required. This is due to vLLM employing numerous custom operators, necessitating the implementation of these same operators for fair comparison purposes.
>
>
> > Question 4: How long does it take to build the first prefix tree? For example, when $n_s=8192$, I think it might take a non-negligible time.
>
> - Let's consider this scenario: with 64 requests in the tree, each having $n_s=8192$, adding a new request ($n_s=8192, s_r=0.5$) takes about 2.5ms. I find this acceptable as the cost only occurs once when a new request is added.
> - The primary overhead in building the prefix tree arises from tensor copying and transposition. Our use of Torch significantly increases the time for this process. However, by employing our custom operators, this time can be significantly reduced, as confirmed by experiments on vllm.

---

> ### Comment · Area_Chair_j8fz · 2023-12-04
> **[Important] Response Required to Authors' Rebuttal**
>
> Dear Reviewer oC3j,
>
> As we progress through the review process for ICLR 2024, I would like to remind you of the importance of the rebuttal phase. The authors have submitted their rebuttals, and it is now imperative for you to engage in this critical aspect of the review process.
>
> Please ensure that you read the authors' responses carefully and provide a thoughtful and constructive follow-up. Your feedback is not only essential for the decision-making process but also invaluable for the authors.
>
> Thank you,
>
> ICLR 2024 Area Chair

---

> > ### Comment · Reviewer_oC3j · 2023-12-04
> > **Response to AC and Authors**
> >
> > I checked the authors' response when it was posted, but it's my fault for not ack.
> > My simple questions are resolved, but I believe the paper can be much more improved when it has more experiments (related to the weaknesses that I stated).
> > Therefore,  I decided to keep my original score at this time.

---

### Official Review · Reviewer_ee1U · 2023-10-30

**Soundness:** 3 good
**Presentation:** 3 good
**Contribution:** 2 fair
**Rating:** 5
**Confidence:** 3

**Summary:**

This paper proposes a new attention kernel design, ChunkAttention, that can chunk, share the KV cache, and batch the attention computation for a multi-tenant LLM inference setting where the sequences from users may share long prompt prefixes. ChunkAttention identifies the matching prompt prefixes across several sequences and reuses their KV cache by chunking the KV cache and embedding it into the auxillary prefix tree.  A 2-phase partitioning technique is also proposed to reduce the memory accesses on the KV cache during self-attention. Experimental results show ChunkAttention improves the self-attention speed by up to 3x.

**Strengths:**

1. The paper works on an important problem, i.e., accelerating self-attention kernels in LLMs.
2. The paper flows well.

**Weaknesses:**

1. The tehniques proposed by this paper work for only the multi-tenant LLM inference setting where the sequences from users may share long prompt prefixes. However, how frequent can this case happen? How to identify the same prompts between different users is still a problem.

2. This authors did NOT compare ChunkAttention agasint prior token pruning techniques, e.g.,

Sehoon Kim, Sheng Shen, David Thorsley, Amir Gholami, Woosuk Kwon, Joseph Hassoun, and
Kurt Keutzer. Learned token pruning for transformers. In Aidong Zhang and Huzefa Rangwala
(eds.), KDD ’22: The 28th ACM SIGKDD Conference on Knowledge Discovery and Data Mining,
Washington, DC, USA, August 14 - 18, 2022, pp. 784–794. ACM, 2022. doi: 10.1145/3534678.
3539260. URL https://doi.org/10.1145/3534678.3539260.

or KV cache compression techniques, e.g.,
Jesse Mu, Xiang Lisa Li, and Noah D. Goodman. Learning to compress prompts with gist tokens.
CoRR, abs/2304.08467, 2023. doi: 10.48550/arXiv.2304.08467. URL https://doi.org/
10.48550/arXiv.2304.08467.

**Questions:**

Please comment on the points in the weakness section.

---

> ### Author Response · Authors · 2023-11-17
>
> 1.	Many LLM applications have system prompts, and system prompts of some important applications like ChatGPT can contain thousands of tokens. Those tokens are shared between all users. We need more time to work out a solution to share the numbers in a compliant way
> 2.	While token pruning technology also aims to improve online inference performance, it works on different aspects. You can apply token pruning together with chunk-attention. We can reference those papers but I don’t think it’s necessary to compare the results

---

### Official Review · Reviewer_sEJt · 2023-10-31

**Soundness:** 3 good
**Presentation:** 3 good
**Contribution:** 4 excellent
**Rating:** 5
**Confidence:** 5

**Summary:**

This paper proposes ChunkAttention, a system optimization to enhance memory reuse within the KV cache in large language model inference workload. In order to boost the efficiency of self-attention for lengthy shared prompts, an efficient computational kernel based on a new tree-based organization of KV cache chunks, which involves the implementation of a two-phase partitioning approach to minimize memory operations on the shared KV cache during self-attention.

**Strengths:**

- Comprehensively, this paper discusses an important system optimization problem in LLM inference, which can introduce significant impact from both academia and industry.

- The design and implementation of the advanced system optimization, especially the tree-based organization of KV-cache chunks is solid and reasonable.

- Most of the technique parts are well written, and easy to follow although it could be further polished.

**Weaknesses:**

- My main concern about the paper is about its evaluation, which can be summarized as below:
  - The organization of this section is unclear; concretely, there is a lack of formal description of the central hypothesis about the experiment design.
  - The setup of the experiments is not clearly stated, I was expecting the experiments would be conducted on some real LLM e.g., Llama2; however, there is no such description.
  - As I stated above, this is a lack of description of the experiment setting; I checked the source code provided in the supplementary materials. Surprisingly, I realized that all the experiments were based on some micro-benchmark on a single transformer layer. (Please correct me if this interpretation is wrong; I am open to discussion, and I can spend more time double-checking the implementation). This makes the experimental results very problematic -- there is no evidence that the IO behavior would be the same when serving a single layer or an end-to-end model. The system should be evaluated for some real workflow instead of this micro-benchmark.

- There is a lack of technique discussion about how this technique can be integrated to parallel inference workflows such as tensor model parallelism or pipeline parallelism.

**Questions:**

The essential question I want to raise is about if it is possible to provide some end-to-end evaluation for some real model over some real LLM inference traces?

---

> ### Author Response · Authors · 2023-11-17
> **The End-to-end Test**
>
> The concern of lacking end-to-end tests on real-world LLM models is valid and well-received. We failed to provide it in the current version due to the following reasons:
> 1. a comparison with existing popular inference solutions, such as vLLM, needs to be done. But these systems all have their highly customized implementations. We need to reach parity on many techniques(noise) unrelated to the optimization covered by this paper, e.g., iteration-based batching, memory offloading, and kernel fusions. Otherwise, the comparison is unfair: where the improvement comes from must be clarified. Is it a result of our optimization or some other techniques? This is also why we only publish micro-kernel level results: they are much less noisy.
> 2. we are working on a highly optimized and production-ready version of this paper's solution. More time is needed before publishing comprehensive end-to-end test results. There will be an updated version of this paper after that.

---

> > ### Comment · Reviewer_sEJt · 2023-12-04
> > **Thank you for your feedback**
> >
> > I really appreciate your feedback. Based on the current status (considering the additional information), I have to keep my score.

---

### Official Review · Reviewer_FRg7 · 2023-10-31

**Soundness:** 2 fair
**Presentation:** 2 fair
**Contribution:** 2 fair
**Rating:** 5
**Confidence:** 3

**Summary:**

The paper aims to tackle the problem of redundant Key-Value (KV) cache in Large Language Models (LLMs) when shared prompt tokens are used. The proposed solution involves the use of a prefix-tree-based data structure for memory management, which can potentially speed up self-attention algorithms.

**Strengths:**

- The paper introduces the concept of using prefix trees for managing KV cache, which is a novel approach in the context of LLMs.

- The problem of addressing redundant KV cache with shared prompt tokens is both relevant and interesting, particularly as LLMs continue to evolve and find application in various fields.

**Weaknesses:**

- The paper lacks support for the need for reducing redundant KV cache through shared prompts. It seems to rely heavily on the assumption that a significant amount of KV cache can be shared, but no data to back this up.

- The paper, particularly Section 3, is difficult to follow. Complex ideas are not explained well, and the method implementation could be better explained to improve the readability.

**Questions:**

- In Section 3.1, the term $n_s$ is used prior to its explanation in later sections, creating a disconnect in the logical flow.

- The paper does not provide any references or empirical evidence to support the concept of sharable KV cache in the context of prompt engineering. This makes it challenging to gauge the validity and impact of the proposed method.

- There is insufficient rationale behind the choice of shared prompt tokens in the experiments. The representativeness of these settings is questionable, and more clarity is needed to assess the paper's contributions effectively.

While the paper tackles an interesting problem and introduces a novel approach via prefix trees, it has substantial limitations. The primary concerns are the lack of empirical data to support the motivation behind reducing redundant KV cache and the complexity in readability, especially in Section 3. These issues, taken together, place the paper below the acceptance threshold for publication in its current form. Revision and additional supporting data are highly recommended.

---

> ### Author Response · Authors · 2023-11-17
>
> 1. Use of Unexplained Terms in Section 3.1: We’ll revise the structure in the next version of the paper to make sure terms are clearly defined before they are used to improve coherence.
>
> 2. References and Empirical Evidence for Sharable KV Cache: The concept of sharable KV cache is an observation from real-world systems, including internal LLM systems and external chatbot plugin systems such as ChatGPT. We are currently in the process of exploring ways to publish empirical data to support this observation, while ensuring we respect all necessary confidentiality constraints.
>
> 3. Rationale Behind Shared Prompt Tokens: The selection of shared prompt tokens in our experiments was made with the intention to cover a broad range of cases observed in prompt systems. These cases can vary widely, with the shared prefix ranging from hundreds to tens of thousands of tokens. In the revised version of our paper, we'll provide a more detailed explanation of our experimental settings to better represent these scenarios.

---

> ### Comment · Area_Chair_j8fz · 2023-12-04
> **[Important] Response Required to Authors' Rebuttal**
>
> Dear Reviewer FRg7,
>
> As we progress through the review process for ICLR 2024, I would like to remind you of the importance of the rebuttal phase. The authors have submitted their rebuttals, and it is now imperative for you to engage in this critical aspect of the review process.
>
> Please ensure that you read the authors' responses carefully and provide a thoughtful and constructive follow-up. Your feedback is not only essential for the decision-making process but also invaluable for the authors.
>
> Thank you,
>
> ICLR 2024 Area Chair

---

> ### Comment · Reviewer_FRg7 · 2023-12-04
>
> Thank the authors for the rebuttal. However, I remain my evaluation on this submission. Gathering more information in the public domain about KVCache sharing and experimental settings could help strengthen the paper.

---

### Meta-Review · Area_Chair_j8fz · 2023-12-06

**Metareview:**

The paper in question introduces a novel concept of using prefix trees for managing KV cache in Large Language Models (LLMs), addressing the issue of redundant KV cache with shared prompt tokens. This approach is both relevant and innovative, particularly in the evolving landscape of LLM applications. However, despite the strengths identified by the reviewers, the paper has significant limitations that lead to an inclination towards rejection, with an average score of 4.5.

While the paper addresses an interesting problem and proposes a novel method, it falls short in crucial areas such as empirical support, clarity, experimental rigor, and comparative analysis. These shortcomings, combined with practical applicability concerns, place the paper below the threshold for acceptance in its current form.

**Justification For Why Not Higher Score:**

The above-combined factors lead to the conclusion that the paper, in its current state, does not meet the threshold for acceptance at the conference.

**Justification For Why Not Lower Score:**

N/A

---

### Decision · Program_Chairs · 2024-01-16

Reject